# sFlt-1/PlGF Ratio Is Not a Good Predictor of Severe COVID-19 nor of Adverse Outcome in Pregnant Women with SARS-CoV-2 Infection—A Case-Control Study

**DOI:** 10.3390/ijerph192215054

**Published:** 2022-11-16

**Authors:** Ewa Malicka, Iwona Szymusik, Beata Rebizant, Filip Dąbrowski, Robert Brawura-Biskupski-Samaha, Katarzyna Kosińska-Kaczyńska

**Affiliations:** Department of Obstetrics, Perinatology and Neonatology, Center of Postgraduate Medical Education, 01-809 Warsaw, Poland

**Keywords:** COVID-19, SARS-CoV-2, pregnancy, outcome, hypertension, preeclampsia

## Abstract

Background: Elevated serum levels of sFlt-1 were found in non-pregnant severe COVID-19 patients. The aim was to investigate sFlt-1/PlGF ratio as a predictor of severe disease and adverse outcome in pregnant women with COVID-19. Methods: A single-center case-control study was conducted in pregnant women with SARS-CoV-2 infection. SARS-CoV-2-negative pregnant women served as controls. Serum sFlt-1/PlGF ratio was assessed. The primary outcome was severe COVID-19 and the secondary outcome comprised adverse outcomes including severe COVID-19, intensive care unit admission, maternal multiple organ failure, preterm delivery, fetal demise, preeclampsia or hypertension diagnosed after COVID-19, maternal death. Results: 138 women with SARS-CoV-2 infection and 140 controls were included. sFlt-1/PlGF ratio was higher in infected patients (11.2 vs. 24; *p* < 0.01) and in women with severe disease (50.8 vs. 16.2; *p* < 0.01). However, it was similar in women with adverse and non-adverse outcome (29.8 vs. 20; *p* = 0.2). The AUC of sFlt-1/PlGF ratio was 0.66 (95% CI 0.56–0.76) for the prediction of severe COVID-19, and 0.72 (95% CI 0.63–0.79) for the prediction of adverse outcome. Conclusions: sFlt-1 and sFlt-1/PlGF ratio are related to SARS-CoV-2 infection and the severity of COVID-19 during pregnancy. However, sFlt-1/PlGF ratio is not a good predictor of severe COVID-19 or adverse outcome.

## 1. Introduction

Coronavirus disease 2019 (COVID-19) is a condition caused by severe acute respiratory syndrome coronavirus 2 (SARS-CoV-2). The World Health Organization (WHO) declared the novel coronavirus outbreak a global pandemic on 11 March 2020 [1]. Since then, it has become one of the leading causes of maternal death worldwide [2]. Studies showed that pregnant women were at an increased risk of severe illness and death compared to non-pregnant ones [3]. According to Jamieson et al. pregnant women infected with SARS-CoV-2 were at a three-fold higher risk of Intensive Care Unit (ICU) admission, 2.9-fold higher risk of mechanical ventilation and 1.7-fold higher risk of death [4].

According to recent knowledge, COVID-19 is characterized by symptoms related to the dysfunction of the renin-angiotensin system (RAS). Renin cleaves angiotensinogen into angiotensin I (ANG I), which is further converted into angiotensin II (ANG II) by Angiotensin-Converting Enzyme-1 (ACE-1). ANG II is then transformed into angiotensin 1-7 (ANG 1-7) by Angiotensin-Converting Enzyme-2 (ACE-2). The activation of the RAS leads to the production of ANG II, which binds type 1 and 2 receptors and functions as a vasoconstrictor, promotes inflammation, coagulation, and fibrosis. SARS-CoV-2 invades cells by binding its spike protein to ACE-2, which is present in the membrane of the alveolar epithelial cells, intestinal epithelial cells, endothelial cells and the trophoblast [4]. RAS components are also present in the trophoblast and contribute to placental invasion, circulation, and angiogenesis during normal pregnancy [5]. Normal pregnancy is characterized by a relative insensitivity to ANG II, allowing low systemic vascular resistance [6].

SARS-CoV-2 binds to ACE-2 and causes its downregulation, leading to the elevation of the level of ANG II. During hypoxia, ANG II promotes the release of soluble fms-like tyrosine kinase 1 (sFlt-1) via binding to type-1 receptor [2,7]. sFlt-1 is a protein associated with endothelial damage, acute lung injury and syncytiotrophoblast hypoxia [8]. It impairs nitric oxide production and makes endothelial cells more sensitive to ANG II, which, in turn, makes them more vulnerable to damage [9]. sFlt-1 and placental growth factor (PlGF) are well known markers of placental hypoxia and predictors of preeclampsia (PE) [10]. Elevated serum levels of sFlt-1 were found in non-pregnant critically ill COVID-19 patients [11]. The above finding suggests its role in COVID-19-associated systemic endothelial dysfunction. So far, several authors have investigated the relationship between sFlt-1/PlGF ratio in pregnant women with SARS-CoV-2 infection and its possible role as a predictor of severe COVID-19. However, their results have been conflicting [2,12,13,14,15]. The study hypothesis assumed that as SARS-CoV-2 affects sFlt-1 release and reflects endothelial damage, women with increased sFlt-1/PlGF ratio would be more prone to severe course of COVID-19. Therefore, the aim of our study was to investigate sFlt-1/PlGF ratio as a predictor of severe disease and adverse outcome in pregnant women with COVID-19.

## 2. Materials and Methods

### 2.1. Study Protocol

This was a single-center retrospective case-control study of pregnant women diagnosed with SARS-CoV-2 infection at the Department of Obstetrics and Gynecology at Bielanski Hospital in Warsaw, Poland. The study was conducted between 1 March 2021 and 31 January 2022. All women had SARS-CoV-2 infection confirmed with an RT-qPCR test. SARS-CoV-2 RT-qPCR-negative pregnant women matched for gestational age, hospitalized at the department within the same period, served as controls. Women were hospitalized because of imminent preterm delivery, preterm rupture of membranes, renal colic, fetal growth restriction, elective cesarean section or labour induction.

The following data were collected from the medical records: age, body mass index (BMI), parity, gestational age at COVID-19 diagnosis, twin pregnancy, preexisting hypertension, gestational diabetes mellitus (GDM), and Doppler assessment (umbilical artery, middle cerebral artery, and uterine artery). COVID-19 respiratory issues such as pneumonia, need for oxygen supplementation, need for continuous positive airway pressure (CPAP) and mechanical ventilation were analyzed. Women with severe COVID-19 were offered high-resolution computed tomography (HRCT). The following biochemical serum test results were collected and analyzed: C-reactive protein (CRP), alanine aminotransferase (ALT), aspartate aminotransferase (AST), creatinine, uric acid, lactate dehydrogenase (LDH), leukocyte count, platelet count, D-dimer, N-terminal prohormone of brain natriuretic peptide (NT-proBNP), placental growth factor (PlGF), sFlt-1, and sFlt-1/PlGF ratio. The following additional data were collected after delivery if the patient was delivered in the same department: hypertension or preeclampsia (PE) diagnosed after SARS-CoV-2 infection, gestational age at delivery, mode of delivery, and neonatal birthweight. Newborns small for gestational age (SGA) were diagnosed if their birthweight was below the 10th percentile for gestational age.

Blood samples were collected on admission from all women. Apart from the routine blood tests conducted according to local policy, plasma concentration of PlGF (Elecsys PlGF, Roche) and sFlt-1 (Elecsys sFlt-1, Roche) were quantified by electrochemiluminescence using an automated analyzer (Cobas-e411, Roche) according to the manufacturer’s instructions. The coefficient of variation for intra-assay for PlGF was 1.6% at 58.9 pg/mL and 2.1% at 284.2 pg/mL and for inter-assay was 5.8% at 58.2 pg/mL and 6.1% at 282.9 pg/mL, while for sFlt-1 was 2.7% at 710 pg/mL and 4.2% at 3910 pg/mL for inter-assay was 7.9% at 760 pg/mL and 7.9% at 3890 pg/mL.

### 2.2. Study Outcomes

The primary outcome was severe COVID-19. It was defined according to NIH as arterial oxygen saturation below 94% on room air, a ratio of arterial partial pressure of oxygen to fraction of inspired oxygen (PaO2/FiO2) <300 mm Hg, a respiratory rate >30 breaths/min, or pulmonary infiltrates >50% [16]. The secondary outcome was adverse outcome including any of the following: severe COVID-19, ICU admission, maternal multiple organ failure, preterm delivery, fetal demise, PE or hypertension diagnosed after COVID-19, or maternal death. Maternal multiple organ failure was diagnosed in case of alteration of two or more organs with a score of ≥3 according to Sequential Organ Failure Assessment Score [17]. Hypertension and PE were diagnosed according to the guidelines of the Polish Society of Obstetricians and Gynecologists [18]. Body mass index (BMI) was calculated by dividing as the actual body mass by the square of the body height.

### 2.3. Statistical Analysis

Variables were described as percentage or median (interquartile range). For statistical analysis, the Mann–Whitney test and the Fisher’s exact test were used. Box-and-whisker plots were created to visualize results. *p*-values < 0.05 were considered significant. Cut-off points were estimated based on ROC curves. The analyzed test was characterized by sensitivity, specificity, positive likelihood ratio and negative likelihood ratio. Logistic regression analysis was performed to investigate the impact of individual factors on primary and secondary outcome. Data were analyzed using Statistica version 13.1, Tibco Software Inc., Palo Alto, CA, USA.

### 2.4. Ethical Issues

The study protocol was approved by the Bioethics Committee at the Center of Postgraduate Medical Education and was conducted in accordance with the World Medical Association Declaration of Helsinki. All enrolled women gave a written informed consent to participate in the study.

## 3. Results

### 3.1. Basic Characteristics of the Study Groups

A total of 165 women diagnosed with SARS-CoV-2 infection were initially included in the study. Twenty-seven were excluded due to the asymptomatic course of infection and no need for hospitalization. Finally, the study group comprised 138 women with SARS-CoV-2 infection. 140 controls were matched for gestational age. In the SARS-CoV-2-positive group 39 women were lost to follow-up after being discharged in good general condition. As regards the control group, 19 women were lost to follow-up. The flow chart of the study groups is presented in Figure 1.

The basic characteristics of the study and control groups are presented in Table 1. There were significantly more obese women in the SARS-CoV-2-positive group. PlGF concentration was similar in both, infected and non-infected groups, while sFlt-1 level was significantly higher in the SARS-CoV-2-positive group. Therefore, sFlt-1/PlGF ratio was also higher in the infected patients. A box and whisker plot of sFlt-1/PlGF serum ratio is presented in Figure 2.

In the COVID-19 group, 19 women developed hypertension or PE within the time span from the infection to delivery, which was significantly more frequent than in the control group (from the time of enrollment to delivery). There were no cases of fetal demise in either the study or control group. Significantly more infected women delivered prematurely, mostly via cesarean section. SARS-CoV-2-positive women gave birth to smaller children. However, the rates of SGA neonates did not differ between the groups. In the infected group, one woman died after having cesarean section at 25 weeks of gestation due to critical respiratory failure, and the baby weighing 680 g died as well within five days.

### 3.2. Primary Outcome: Severe Course of COVID-19

As regards the group of 138 women with SARS-CoV-2 infection, 57 had severe COVID-19 (41.3%). All required oxygen supplementation, including seven via a nasal cannula, nine via an oxygen facial mask, and twenty-three via an oxygen facial mask with reservoir; nine had CPAP applied and nine were mechanically ventilated. Nine patients were admitted to the ICU. Eighteen women underwent HRCT. The mean percentage of affected lung tissue was 40 (interquartile range 20–70).

The comparison of severe and non-severe COVID-19 patients is presented in Table 2. SARS-CoV-2 was diagnosed at similar gestational age. The occurrence of preexisting hypertension or PE was comparable, while women with severe COVID-19 had GDM diagnosed significantly more often. Differences in CRP, ALT, AST and LDH levels were observed between severe and non-severe COVID-19 groups. Women with severe COVID-19 had a higher pulsatility index in the uterine arteries and lower in the fetal middle cerebral arteries. Patients with severe disease had a higher serum level of sFlt-1 and higher sFlt-1/PlGF ratio. A box and whisker plot of sFlt-1/PlGF serum ratio is presented in Figure 2.

The rates of hypertension and PE diagnosed before or after COVID-19 were similar in both groups. The rates of preterm deliveries and small for gestational age neonates were similar, while significantly more women with severe COVID-19 delivered via cesarean section.

In 19 women, SARS-CoV-2 infection was diagnosed at or below 26 weeks of gestation. None of them had preexisting hypertension, while four developed hypertension and three were diagnosed with PE following COVID-19. Eight women suffered from severe disease. Ten women were discharged and lost to follow-up. Nine women delivered, seven within three weeks after COVID-19 diagnosis due to critical illness, and two had cesarean section performed at 32 and 34 weeks of gestation due to severe PE.

### 3.3. Secondary Outcome: Adverse Outcomes

Seventy-five women with COVID-19 had adverse outcomes: 57 had severe COVID-19, nine were hospitalized in the ICU. There were no cases of maternal multiple organ failure nor fetal demise, one woman died of respiratory insufficiency (as mentioned above), 19 developed hypertension or PE after COVID-19 and 27 delivered prematurely. The comparative characteristics of women with and without adverse outcomes are presented in Table 3. As regards women who presented adverse outcome, COVID-19 was diagnosed significantly earlier in pregnancy. They had higher levels of CRP, ALT, LDH, NT-proBNP, higher pulsatility index in the uterine arteries and lower in the fetal middle cerebral arteries in Doppler assessment. Significantly more women from the adverse outcome group delivered prematurely. Their newborns had significantly lower birthweight and were more often small for gestational age. sFlt-1/PlGF ratio was similar in women with adverse and non-adverse outcome.

The analysis of COVID-19 prevalence at specific weeks of pregnancy is presented in Figure 3. There were two peaks of morbidity: 24 to 26 weeks and >32 weeks of gestation. Of the women, 50.9% with severe COVID-19 became infected >36 weeks of gestation. Adverse outcomes occurred mostly between 23 and 26 weeks or >32 weeks of gestation.

### 3.4. Association between Biochemical Markers and Study Outcomes

The biochemical markers analyzed regarding the association with study outcomes were as follows: CRP, ALT, AST, creatinine, uric acid, LDH, leucocyte count, platelet count, D-dimer, NT-proBNP and sFlt-1/PlGF ratio. The only ones associated with severe COVID-19 included CRP (adjusted odds ratio (aOR) 1.04, 95% confidence interval (CI) 1.02–1.06; *p* < 0.01) and sFlt-1/PlGF ratio (aOR 1.04, 95% CI 1.02–1.05; *p* < 0.01). The AUC of sFlt-1/PlGF ratio for the prediction of severe COVID-19 was 0.66 (95% CI 0.56–0.76). The ROC curve is presented in Figure 4. The detection rate at 5% false-positive rate was 26.3% and 40.3% at 10% false-positive rate. The best cut-off value of sFlt-1/PlGF ratio for severe COVID-19 prediction was 46.3, with a sensitivity of 0.54 (95% CI 0.4–0.8), specificity of 0.85 (95% CI 0.8–0.9), positive likelihood ratio of 3.67 (95% CI 2.1–6.5) and negative likelihood ratio of 0.54 (0.4–0.7).

An analogous analysis was performed for adverse outcome. As regards the analyzed variables, only CRP (aOR 1.03, 95% CI 1.01–1.04; *p* < 0.01) and sFlt-1/PlGF ratio (aOR 1.02, 95% CI 1–1.03; *p* = 0.04) were associated with adverse outcome. The AUC of sFlt-1/PlGF ratio for the prediction of adverse outcome was 0.72 (95% CI 0.63–0.79). The ROC curve is presented in Figure 4. The detection rate at 5% false-positive rate was 21.3%, and at the 10% false-positive rate it reached 40%. The best cut-off value for severe COVID-19 prediction was 46.3 with a sensitivity of 0.45 (95% CI 0.3–0.6), specificity of 0.86 (95% CI 0.8–0.9), positive likelihood ratio of 3.17 (95% CI 1.7–6.1) and negative likelihood ratio of 0.64 (0.5–0.8).

## 4. Discussion

We found significant differences in sFlt-1 and sFlt-1/PlGF ratio between SARS-CoV-2-negative and positive pregnant women, as well as in patients with non-severe and severe COVID-19. However, sFlt-1/PlGF ratio was not a good predictor of severe COVID-19, nor adverse outcome.

The effect of COVID-19 on the course of pregnancy was investigated in a large, longitudinal, prospective study by Papageorghiou et al. The study enrolled 2184 pregnant women. COVID-19 was diagnosed in 725 (33.2%) of them. After adjustment for sociodemographic factors and conditions associated with both COVID-19 and PE, the risk ratio for PE was significantly greater in COVID-19 group (risk ratio (RR) 1.77, 95% CI 1.25–2.52) [19]. A recent meta-analysis revealed SARS-CoV-2 infection during pregnancy to be associated with a significant increase in the odds of developing PE (OR 1.58, 95% CI 1.39–1.8), PE with severe features (OR 1.76, 95% CI 1.18–2.63) and eclampsia (OR 1.97, 95% CI 1.01–3.84) [20]. A correlation between the severity of infection and PE risk was also observed. Lai et al. found severe COVID-19 disease to be associated with a higher risk of PE (adjusted RR 4.9, 95% CI 1.56–15.38). The risk was lower (adjusted RR 3.3, 95% CI 1.48–7.38) in patients with moderate COVID-19 compared to those with asymptomatic or mild disease [21]. Our study revealed that subsequent hypertension or PE was more prevalent in the COVID-19 group. Endothelial dysfunction caused by SARS-CoV-2 is one of the possible mechanisms of PE development in affected women. Nowadays, the virus has been recognized as one of the possible etiological factors of PE development [22]. According to this hypothesis, SARS-CoV-2 binds to ACE-2 and promotes ANG II, blocking its transformation into ANG 1–7. As ANG II promotes the release of sFlt-1, it is hypothesized that the virus induces sFlt-1 increase in the blood. Giardini et al. found significant differences in sFlt-1/PlGF ratio between non-pregnant patients with and without COVID-19 [12]. The correlation between sFlt-1 and SARS-CoV-2 infection in pregnant women has been investigated as well. However, research data published on this subject so far have been conflicting. A study by Espino-y-Sosa et al. demonstrated that pregnant women with severe COVID-19 had an elevated maternal plasma concentration of sFlt-1 and a high sFlt-1/PlGF ratio [2]. Significant differences were observed in our study between SARS-CoV-2-positive and negative women and between severe and non-severe COVID-19 patients as well. In our cohort, women with SARS-CoV-2 infection had a higher serum sFlt-1/PlGF ratio than non-infected ones. Women with severe COVID-19 also had a higher sFlt-1/PlGF ratio than patients with mild disease. Soldavini et al. analyzed a cohort of pregnant women affected by COVID-19. They found higher prevalence of hypertensive disorders in pregnancy in women affected by COVID-19. However, the authors did not find sFlt-1/PlGF ratio to be significantly higher in the group with COVID-19 compared to those who did not develop the disease. The ratio was elevated in hypertensive women, with and without SARS-CoV-2 infection. Significant differences were observed between COVID-19 hypertensive and normotensive women and between non-COVID-19 hypertensive and COVID-19 normotensive patients. COVID-19 did not deteriorate the angiogenic imbalance in pregnant women [9].

sFlt-1/PlGF ratio has been evaluated as a predictor of severe COVID-19. Data published on this subject are also conflicting. In the majority of studies, the correlation between the ratio and the course of infection was observed. Espino-y-Sosa et al. investigated 80 pregnant women with COVID-19 and found sFlt-1/PlGF ratio to be significantly higher in patients with severe illness. The authors performed multiple logistic regression and found that a high sFlt-1/PlGF ratio was associated with severe COVID-19 (OR 1.02, 95% CI 1–1.03; *p* = 0.002) [2]. Their results are similar to ours. We also found that sFlt-1/PlGF ratio was related to severe COVID-19 (aOR 1.04, 95% CI 1.02–1.05; *p* < 0.01). Torres-Torres et al. investigated a cohort of 113 pregnant women with COVID-19 and discovered the association of biochemical markers and the severity of infection during pregnancy. They found similar levels of serum PlGF in patients with non-severe and severe pneumonia and significantly higher concentrations of sFlt-1 in severe pneumonia patients [14]. Their observations are also in accordance with our results. The levels of PlGF were similar in controls and infected women, as well as in severe and non-severe COVID-19 patients, while sFlt-1 concentrations were significantly higher in infected and severe COVID-19 women. Torres-Torres at al. converted sFlt-1 measurements into multiples of median (MoM) and found sFlt-1 MoM to be significantly higher in severe pneumonia group (1.81, 95% CI 0.75–5.3 **vs.** 0.76, 95% CI 0.54–1.38; *p* = 0.001). Their study showed that higher sFlt-1 MoM was associated with an increased risk of severe pneumonia (aOR 1.817, 95% CI 1.365–2.418), ICU admission (aOR 2.195, 95% CI 1.582–3.047), viral sepsis (aOR 2.318, 95% CI 1.407–3.820) and maternal death (unadjusted OR 5.504, 95% CI, 1.079–28.076) [14]. In our study, sFlt-1/PlGF ratio was associated with severe COVID-19 and adverse outcomes as well. Conversely, no differences in sFlt-1/PlGF ratio between pregnant women with asymptomatic and symptomatic SARS-CoV-2 infection were observed by Giardini et al. The authors analyzed a group of 37 asymptomatic women and 20 patients with symptoms of COVID-19. They found lower levels of sFlt-1 in symptomatic women, while PlGF concentrations were similar in both groups [13]. The results published by Giardini et al. were opposite to those previously described by Esspino-y Sosa et al. and Torres-Torres et al., as well as to those presented in our study. The observed differences may be due to small sample groups in a study by Giardini et al., as the authors investigated only 20 women with a symptomatic SARS-CoV-2 infection. Higher gestational age in asymptomatic patients could also introduce bias, as sFlt-1 increases with gestational age [23]. In our study, 138 women infected with SARS-CoV-2 were included, with 57 presenting severe COVID-19 symptoms. The controls were matched for gestational age. The time between the collection of blood sample and the first symptoms of COVID-19 is another factor that can influence the results. All blood samples were collected during hospital admission in our study.

Other components of the RAS were investigated in pregnant women with COVID-19. Espino-y-Sosa et al. evaluated ACE-2, ANG II and sFlt-1 serum levels in 80 pregnant women with SARS-CoV-2 infection. The authors found no differences in ACE-2 concentrations between patients with non-severe and severe COVID-19, while the serum levels of ANG II were higher in women with severe illness. A significant correlation was identified between sFlt-1 and ANG II for severe pneumonia [2]. The authors calculated sFlt-1/ANG II ratio and found its significant association with severe pneumonia (OR 1.31, 95% CI 1.09–1.56, *p* = 0.003). sFlt-1/ANG II ratio was found to be a better predictor of adverse outcomes in COVID-19. Its AUC was 0.96 (95% CI 0.807–0.981) with the detection rates of 52% at 5% false-positive rate and 88% at 10% false-positive rate [2].

The main strength of our study is that a homogenous cohort of women were diagnosed and investigated at a single center. All women had an RT-qPCR test performed, including the asymptomatic pregnant women in the control group. The control group was matched for gestational age with the SARS-CoV-2-positive group. All infected women were treated according to the same local policy. We collected blood samples on admission, not during the whole period of hospitalization. The assays of sFlt-1 and PlGF were performed according to the instructions of Roche Diagnostics. However, the study has some limitations. First, it was a retrospective analysis of the medical data of women hospitalized in Bielanski Hospital. The study population could, therefore, be highly selected because of the nature of a hospital-based observational study, especially in the tertiary perinatal center. There was only one case of maternal death and no fetal demise in the study group. Thirty-nine women with SARS-CoV-2 infection were lost to follow-up after recovery and data on post-COVID complications and delivery were available for 99 women.

## 5. Conclusions

sFlt-1 and sFlt-1/PlGF ratio are related to SARS-CoV-2 infection and the severity of COVID-19 during pregnancy. However, sFlt-1/PlGF ratio is not a good predictor of severe COVID-19 or adverse outcome. More prospective large studies are required to investigate the RAS in SARS-CoV-2 infection during pregnancy and to evaluate which biochemical markers are the best predictors of COVID-19 severity in pregnant women.

## Figures and Tables

**Figure 1 ijerph-19-15054-f001:**
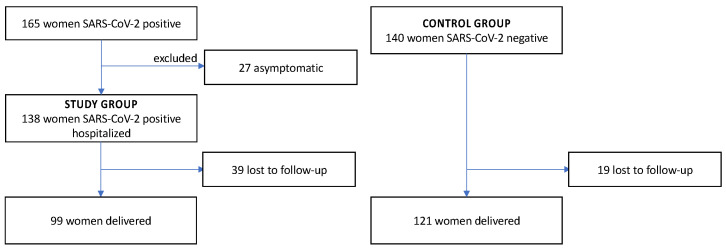
The flow chart of the study groups.

**Figure 2 ijerph-19-15054-f002:**
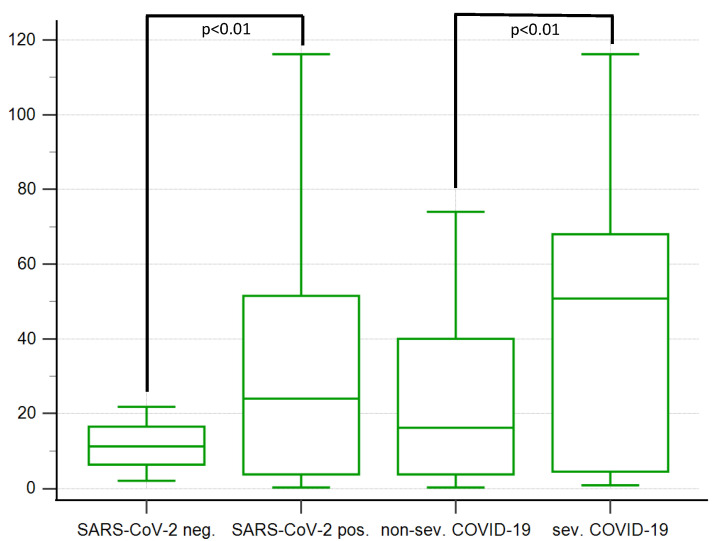
A box and whisker plot of sFlt-1/PlGF serum ratio in SARS-CoV-2-negative and positive, non-severe and severe COVID-19 groups. SARS-CoV-2-neg.—SARS-CoV-2-negative; SARS-CoV-2-pos.—SARS-CoV-2-positive; non-sev. COVID-19—non-severe COVID-19; sev. COVID-19—severe COVID-19.

**Figure 3 ijerph-19-15054-f003:**
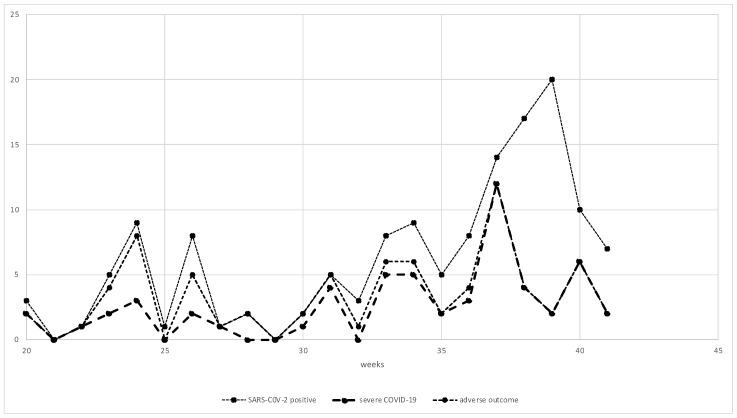
COVID-19 frequency in particular weeks of pregnancy.

**Figure 4 ijerph-19-15054-f004:**
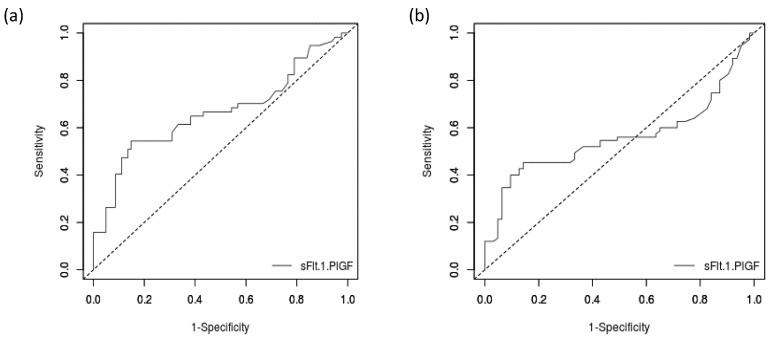
ROC curves for sFlt-1/PlGF in the prediction of study outcomes. (**a**) ROC curve for sFlt-1/PlGF in the prediction of severe COVID-19. (**b**) ROC curve for sFlt-1/PlGF in the prediction of adverse outcomes.

**Table 1 ijerph-19-15054-t001:** Basic characteristics of the study and control groups.

	SARS-CoV-2-Positive	Controls	
	N = 138	N = 140	
	N (%)	N (%)	*p*
age (years) *	32 (28–36)	32 (27–36)	0.9
age ≥ 35	37 (26.8)	39 (27.8)	0.9
BMI on admission (kg/m^2^) *	31 (25–34)	28 (23–31)	0.04
BMI ≥ 30	47 (34.1)	16 (11.4)	<0.01
nulliparous	54 (39.1)	62 (44.3)	0.4
gestational age at COVID-19 diagnosis (weeks) *	36 (31–39)	--------	
twin pregnancy	12 (8.7)	5 (3.6)	0.08
previous hypertension or PE	6 (4.3)	7 (5)	0.8
GDM	16 (11.6)	13 (9.3)	0.5
PlGF (pg/mL) *	154.4 (55.2–271.8)	146.7 (64.9–204.1)	0.7
sFlt-1 (pg/mL) *	2617 (1010–5100)	1112 (710–1615)	<0.01
sFlt-1/PlGF ratio *	24 (3.7–51.5)	11.2 (2.1–23.4)	<0.01
delivery	N = 99	N = 121	
subsequent hypertension or PE	19 (19.2)	2 (1.7)	<0.01
gestational age at delivery (weeks) *	37 (34–39)	39 (36–42)	0.1
PTD	31 (31.3)	8 (6.6)	<0.01
cesarean section	70 (70.7)	56 (56.6)	<0.01
neonatal birthweight (g) *	3130 (2542–3680)	3420 (2880–3920)	0.01
birthweight <10 pc	11 (11.1)	10 (8.3)	0.5

* median (interquartile range). BMI—body mass index; GDM—gestational diabetes mellitus; PlGF—placental growth factor; sFlt-1—soluble fms-like tyrosine kinase 1; PE—preeclampsia; PTD—preterm delivery (<37 weeks of gestation).

**Table 2 ijerph-19-15054-t002:** Characteristics of severe and non-severe COVID-19 patients.

	Non-Severe COVID-19	Severe COVID-19	
	N = 81	N = 57	
	N (%)	N (%)	*p*
age (years) *	32 (27–37)	33 (29–34)	0.8
BMI on admission (kg/m^2^) *	29 (23–33)	33 (26–38)	0.2
BMI ≥ 30	28 (34.5)	19 (33.3)	0.7
nulliparous	28 (34.6)	26 (45.6)	0.2
gestational age at COVID-19 diagnosis (weeks) *	37 (30–39)	36 (31–37.2)	0.3
twin pregnancy	7 (8.6)	5 (8.8)	0.9
previous hypertension or PE	2 (2.5)	4 (7)	0.2
GDM	0	16 (28.1)	<0.01
UtA PI	0.7 (0.5–0.9)	0.9 (0.7–1.2)	0.04
UA PI	0.8 (0.7–1)	0.9 (0.8–1)	0.2
MCA PI	1.5 (1.4–2)	1.4 (1–1.6)	0.04
CRP (mg/L) *	10.3 (5.4–15.8)	37 (11.3–116.9)	<0.01
ALT (U/L) *	16.4 (10.3–30.7)	26.7 (12.6–76.3)	0.04
AST (U/L) *	19.9 (15.4–32.1)	32.2 (16.2–71.3)	0.004
creatinine (mg/dL) *	0.5 (0.5–0.6)	0.6 (0.5–0.7)	0.08
uric acid (mg/dL) *	4.3 (3.9–5.6)	4.8 (4–5.9)	0.4
LDH (U/L) *	179.1 (153–203)	228 (177.5–297.5)	<0.01
WBC (×10^9^/L) *	9.5 (7–11.2)	8.8 (6.7–11.4)	0.8
PLT (×10^3^/L) *	193 (150–251)	213 (146–319)	0.2
D-dimer (ng/mL) *	2.5 (1.3–3)	1.9 (1–3.4)	0.4
NT-proBNP (pg/mL) *	44.8 (30–92.7)	50.2 (30.3–142.7)	0.4
PlGF (pg/mL) *	161.9 (53.2–289)	148.4 (51.8–264.8)	0.5
sFlt-1 (pg/mL) *	2290 (980–4890)	3620 (1880–9100)	<0.01
sFlt-1/PlGF ratio	16.2 (3.7–40)	50.8 (4.5–68)	<0.01
delivered	N = 58	N = 41	
post-COVID hypertension or PE	8 (13.8)	11 (26.8)	0.1
gestational age at delivery (weeks) *	38 (36–39)	37 (34.8–38.3)	0.1
PTD	13 (22.4)	18 (43.9)	0.1
cesarean section	32 (55.2)	38 (92.7)	<0.01
neonatal birthweight (g) *	3270 (2810–3720)	3070 (2460–3475)	0.09
birthweight <10 pc	8 (13.8)	3 (7.3)	0.05

* median (interquartile range). BMI—body mass index; GDM—gestational diabetes mellitus; UtA PI—uterine artery pulsatility index; UA PI—umbilical artery pulsatility index; MCA PI—middle cerebral artery pulsatility index; CRP—C-reactive protein; ALT—alanine aminotransferase; AST—aspartate aminotransferase; LDH—lactate dehydrogenase; WBC—leukocyte count; PLT—platelet count; NT-proBNP—N-terminal prohormone of brain natriuretic peptide; PlGF—placental growth factor; sFlt-1—soluble fms-like tyrosine kinase 1; PE—preeclampsia; PTD—preterm delivery (<37 weeks of gestation).

**Table 3 ijerph-19-15054-t003:** Characteristics of women with and without adverse outcomes.

	Non-Adverse Outcome	Adverse Outcome	
	N = 63	N = 75	
	N (%)	N (%)	*p*
age (years) *	31.5 (27.3–37)	33 (29–34)	0.9
BMI on admission (kg/m^2^) *	31 (24–34)	32 (25–37)	0.8
BMI ≥ 30	23 (36.5)	24 (32)	0.5
nulliparous	27 (42.9)	27 (36)	0.4
gestational age at COVID-19 diagnosis (weeks) *	37 (35–39)	34 (26–37)	<0.01
twin pregnancy	3 (4.8)	9 (12)	0.2
previous hypertension or PE	2 (3.2)	4 (5.3)	0.7
GDM	1 (1.6)	15 (20)	<0.01
UtA PI	0.7 (0.5–0.9)	1 (0.8–1.2)	0.02
UA PI	0.7 (0.6–0.9)	0.9 (0.8–1)	0.1
MCA PI	1.5 (1.3–2)	1.3 (0.9–1.5)	0.02
CRP (mg/L) *	10.3 (5.6–16.3)	31.5 (10.1–90.2)	<0.01
ALT (U/L) *	15.4 (10.1–27.2)	20.2 (13.2–63)	0.04
AST (U/L) *	20 (15.6–32.1)	27.8 (15.9–62.3)	0.09
creatinine (mg/dL) *	0.5 (0.5–0.6)	0.6 (0.5–0.7)	0.5
uric acid (mg/dL) *	5 (3.9–5.6)	4.6 (3.9–5.9)	0.9
LDH (U/L) *	179 (154.8–202.3)	204 (164–289.3)	<0.01
WBC (×10^9^/L) *	9.5 (6.7–11.2)	8.9 (7.2–12.1)	0.6
PLT (×10^3^/L) *	205 (151.5–253.3)	211 (145.3–262.3)	0.7
D-dimer (ng/mL) *	2.5 (1.3–2.9)	1.9 (1–3.4)	0.8
NT-proBNP (pg/mL) *	42.9 (25.8–69.4)	60.3 (28.6–149)	0.02
PlGF (pg/mL) *	148.1 (50.8–267)	153.3 (53.8–289.2)	0.3
sFlt-1 (pg/mL) *	2411 (1005–4980)	2648 (1045–5130)	0.6
sFlt-1/PlGF ratio	20 (5–41)	29.8 (2.7–63.8)	0.2
delivered	N = 48	N = 51	
post-COVID hypertension or PE	0 (8.3)	19 (37.3)	<0.01
gestational age at delivery (weeks) *	39 (38–39)	37 (34–38)	<0.01
PTD	4 (16.7)	27 (45.1)	<0.01
cesarean section	32 (66.7)	38 (74.5)	0.4
neonatal birthweight (g) *	3475 (3090–3815)	2810 (2207–3300)	<0.01
birthweight < 10 pc	2 (4.2)	9 (17.6)	0.03

* median (interquartile range). BMI—body mass index; GDM—gestational diabetes mellitus; UtA PI—uterine artery pulsatility index; UA PI—umbilical artery pulsatility index; MCA PI—middle cerebral artery pulsatility index; CRP—C-reactive protein; ALT—alanine aminotransferase; AST—aspartate aminotransferase; LDH—lactate dehydrogenase; WBC—leukocyte count; PLT—platelet count; NT-proBNP—N-terminal prohormone of brain natriuretic peptide; PlGF—placental growth factor; sFlt-1—soluble fms-like tyrosine kinase 1; PE—preeclampsia; PTD—preterm delivery (<37 weeks of gestation).

## Data Availability

Data are available on request.

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
