# Peer review of "sFlt-1/PlGF Ratio Is Not a Good Predictor of Severe COVID-19 nor of Adverse Outcome in Pregnant Women with SARS-CoV-2 Infection—A Case-Control Study"

_ijerph, 2022, doi:10.3390/ijerph192215054_

Round 1
Reviewer 1 Report
Comments for authors are in the attached file.

Author Response
Dear Editor,
I thank the Reviewer for the constructive comment. I have revised the manuscript according to the suggestions and I hope that the changes will convince the Reviewer and the Editor that the paper is worthy of publication in International Journal of Environmental Research and Public Health.
Below are the answers to the Reviewer’s suggestions:
- In the Flow chart on page 3, Starting off the chart as two different groups e.g., control and infected would offer better visualization and easy reading.
The figure was changed for better visualization of the study and the control group
- Table 1. Please correct “B_M_I_ _(_k_m_/_m_2)_” _to “(kg/m2)”.
The typo has been corrected.
- In the methods section, give subtitles to the methods used
The subtitles have been added.
- Mention the inter assay and intra assay coefficient of variability for the sFLT-1 and PLGF ELISA.
The inter and intra assays have been added to the manuscript.
Reviewer 2 Report
In the interesting manuscript 'sFlt-1/PlGF ratio is not a good predictor of severe COVID-19 nor of adverse outcome in pregnant women with SARS-CoV-2 infection – a case-control study' the authors evaluated circulating angiogenic (preeclampsia) marker proteins in maternal blood in pregnant women suffering from COVID-19. They aimed at the predictive potential for disease severity and adverse outcome. This is a monocentric study of 138 women suffered from COVID-19 and 140 control women without SARS-CoV-2 infection. The authors stated that they analysed medical data retrospectively. sFlt-1 and PlGF was measured in both groups. sFlt-1 / PlGF ratio was higher in the COVID-19 group compared to the control group, as was sFlt-1 concentration alone. However, predictive value for severe COVID-19 and defined adverse outcomes was low.
The scientific rationale for the study is clear. sFlt-1/PlGF ratios are maternal blood markers that have been shown to predict preeclampsia and preeclampsia-related adverse outcomes. Because COVID-19 has been hypothesized to be a risk factor for preeclampsia and to share pathways to endothelial dysfunction, it could be argued that angiogenic factors such as sFlt-1 and PlGF also play a role in COVID-19 progression of illness and are therefore predictive of adverse outcomes.
The paper over all is well written however, there are some major concerns with the study design that need to be addressed:
1. The authors stated that they wanted to investigate the sFlt-1/PlGF ratio as a predictor of severe COVID-19. However, the predictive value can be meaningfully determined only in the early phase of infection. Because the authors did not specify a specific time point for measuring the sFlt-1/PlGF ratio, it must be assumed that severe COVID-19 infection was already present when the blood was drawn.
2. In addition, of the 165 women diagnosed with SARS-CoV-2 infection, 57 (34.5%) developed severe COVID-19. Therefore, this population is highly selected because of the nature of a hospital-based observational study. It is very likely that most of the 57 women with severe COVID-19 were already severely ill when they were admitted to the hospital.
3. The authors should also consider whether women with a higher degree of endothelial or placental dysfunction before or on infection with SARS-CoV-2 are more likely to develop severe symptoms and are therefore more likely to be hospitalized. Thus, this means that among women with severe COVID-19, one finds more often women with preeclampsia and women with a higher sFlt-1/PlGF ratio than in a control group, but this rather reflects the health status of the women before infection with SARS-CoV-2 and has nothing to do with the SARS-CoV-2 infection. The question therefore arises whether a higher sFlt-1/PlGF ratio is correlated with disease severity by chance or by selection of the population, respectively. This consideration is underlined by the higher PI in uterine arteries in the COVID-19 group. Higher uterine artery PI is certainly not related to COVID-19 but to failed trophoblast invasion occurring early in pregnancy and thus weeks before infection.
4. In the methods, authors should clearly state whether this is a prospective or retrospective study.
5. in the methods, the authors need to describe more clearly how they included the control group. For what reason did the women in the control group visit the hospital? Was this a group of women with uncomplicated pregnancies who visited the hospital for routine antenatal care?
6. Is sFlt-1/PlGF routinely measured in all women admitted to the hospital?
7. It is also not clear, when during gestational age blood was drawn and analyzed neither in the COVID-19 group nor in the control group. Without this information, analysis is meaningless. (table 1).
8. Authors should provide a proper hypothesis in the introduction.
9. Authors should indicate in the tables whether they used prepregnancy BMI or BMI at admission.
10. C-section rate is very high (56% in the control group). The authors should state on the high incidence of c-section. This is rather indicative for a highly selected high-risk group.
11. Table 3. Page 8. sFlt-1 24110 (here a '0' was added by mistake)
12. Overall, i) it is unclear to the reviewer whether the sFlt-1/PlGF ratio is coincidentally correlated with severe COVID-19 and not causative, and ii) the study design does not allow analyzing the predictive value of sFlt-1/PlGF
Author Response
Dear Editor,
I thank the Reviewer for the astute comment. I have revised the manuscript according to the suggestions and I hope that the changes will convince the Reviewer and the Editor that the paper is worthy of publication in International Journal of Environmental Research and Public Health.
Below are the answers to the Reviewer’s suggestions:
- The authors stated that they wanted to investigate the sFlt-1/PlGF ratio as a predictor of severe COVID-19. However, the predictive value can be meaningfully determined only in the early phase of infection. Because the authors did not specify a specific time point for measuring the sFlt-1/PlGF ratio, it must be assumed that severe COVID-19 infection was already present when the blood was drawn.
Blood samples were collected on admission from all women. This information is included in the manuscript. Admission to hospital was the only established moment during the disease. We are aware that it is not possible to specify the time of infection before admission nor to design a study in which blood samples could be collected in the same moment in every woman as it is not possible to evaluate the moment of infection of SARS-CoV-2. However, the study was designed to evaluate if sFlt-1/PlGF ratio was a predictor of severe disease and adverse outcome in pregnant women diagnosed with COVID-19.
- In addition, of the 165 women diagnosed with SARS-CoV-2 infection, 57 (34.5%) developed severe COVID-19. Therefore, this population is highly selected because of the nature of a hospital-based observational study. It is very likely that most of the 57 women with severe COVID-19 were already severely ill when they were admitted to the hospital.
We are aware of the bias introduced by specificity of the study group. However, as mentioned earlier, the study was designed to evaluate if sFlt-1/PlGF ratio was a predictor of severe disease and adverse outcome in pregnant women diagnosed with COVID-19 and in this way is valuable for clinical practice. The limitation of the study has been added to the manuscript.
- The authors should also consider whether women with a higher degree of endothelial or placental dysfunction before or on infection with SARS-CoV-2 are more likely to develop severe symptoms and are therefore more likely to be hospitalized. Thus, this means that among women with severe COVID-19, one finds more often women with preeclampsia and women with a higher sFlt-1/PlGF ratio than in a control group, but this rather reflects the health status of the women before infection with SARS-CoV-2 and has nothing to do with the SARS-CoV-2 infection. The question therefore arises whether a higher sFlt-1/PlGF ratio is correlated with disease severity by chance or by selection of the population, respectively. This consideration is underlined by the higher PI in uterine arteries in the COVID-19 group. Higher uterine artery PI is certainly not related to COVID-19 but to failed trophoblast invasion occurring early in pregnancy and thus weeks before infection.
Thank you for this astute comment. Indeed, we could not evaluate which women had higher sFlt-1/PlGF ratio before the SARS-CoV-2 infection. It is possible that preexisting endothelial damage can influence the severity of COVID-19. This way sFlt-1/PlGF could still be useful in the prognosis of severe disease or adverse outcome. To investigate it another study should be conducted. It should include women with known sFlt-1/PlGF ratio observed before SARS-Cov-2 infection. In our study it is not possible to verify this hypothesis.
- In the methods, authors should clearly state whether this is a prospective or retrospective study.
It was a retrospective study and this information was added to the manuscript.
- in the methods, the authors need to describe more clearly how they included the control group. For what reason did the women in the control group visit the hospital? Was this a group of women with uncomplicated pregnancies who visited the hospital for routine antenatal care?
The control group consisted of pregnant women matched for gestational age, hospitalized at the department within the same period. Women were hospitalized because of imminent preterm delivery, preterm rupture of membranes, renal colic, fetal growth restriction, elective cesarean section or labour induction. Those details were added to the manuscript.
- Is sFlt-1/PlGF routinely measured in all women admitted to the hospital?
During the study period all women had sFlt-1/PlGF ratio measured.
- It is also not clear, when during gestational age blood was drawn and analyzed neither in the COVID-19 group nor in the control group. Without this information, analysis is meaningless. (table 1).
The median gestational age at COVID-19 diagnosis was 36 weeks in the study group. The blood samples were collected at diagnosis and hospital admission. The control group was matched for gestational age.
- Authors should provide a proper hypothesis in the introduction.
A The study hypothesis has been added to the manuscript.
- Authors should indicate in the tables whether they used prepregnancy BMI or BMI at admission.
We used the value calculated from the body weight at admission. This information has been added to the text and the tables.
- C-section rate is very high (56% in the control group). The authors should state on the high incidence of c-section. This is rather indicative for a highly selected high-risk group.
The cesarean section rate in the control group was similar to the general rate in our Department as it is a tertiary perinatal center. The rate of cesarean deliveries in Poland is 47%. It was commented on as a limitation of the study.
- Table 3. Page 8. sFlt-1 24110 (here a '0' was added by mistake)
The typo was corrected.
- Overall, i) it is unclear to the reviewer whether the sFlt-1/PlGF ratio is coincidentally correlated with severe COVID-19 and not causative, and ii) the study design does not allow analyzing the predictive value of sFlt-1/PlGF
Thank you for this comment. Basing on our study it was not possible to determine if sFlt-1/PlGF ratio is a cause of severe COVID-19 OR coincidentally correlated with it. However, it was possible to investigate the predictive value of sFlt-1/PlGF, measured on admission, in prediction of severe course of disease in hospitalized pregnant women.